# The Resource Status of Plankton after Stocked *Protosalanx chinensis* Population Collapse in a Lake of Northeastern China

**Zhe Li** [1,2,3,4], **Ying Han** [1,*], **Fujiang Tang** [2,3,4,*], **Haoyu Zeng** [2,3,4] and **Yi Zheng** [2,3,4]

[1] College of Animal Science and Technology, Northeast Agricultural University, No. 600, Changjiang Road, Xiangfang District, Harbin 150030, China; lizhe@hrfri.ac.cn
[2] Heilongjiang River Fisheries Research Institute, Chinese Academy of Fishery Sciences, Harbin 150070, China
[3] Scientific Observing and Experimental Station of Fishery Resources and Environment in Heilongjiang River Basin, Ministry of Agriculture and Rural Affairs, Harbin 150070, China
[4] National Agricultural Experimental Station for Fishery Resources and Environment, Fuyuan 156500, China
[*] Correspondence: hanyingzou@neau.edu.cn (Y.H.); rivery2008@163.com (F.T.)

**Abstract:** In order to understand the relative strength of top-down and bottom-up control in lakes of grazing alien fish, *Protosalanx chinensis,* investigations were designed in Lake Longhu (Lake L, *P. chinensis* introduced in 2013) and Lake Qijia (Lake Q, *P. chinensis* free). Plankton samples were collected bimonthly through the whole life cycle of *P. chinensis* (from February to December in 2018). A total of 133 phytoplankton and 68 zooplankton species were recorded in the two lakes. The total density and biomass of phytoplankton and zooplankton were $991.89 \times 10^4$ ind/L and 9.2418 mg/L as well as 5212 ind/L and 20.2646 mg/L, respectively. This study revealed that *P. chinensis* grazing did not deplete the zooplankton resources in the lake where it was stocked. Biodiversity in Lake L was not significantly different from that in Lake Q based on both phytoplankton and zooplankton. Overall, the over grazing of *P. chinensis* was not found in Lake L. Nevertheless, compared to Lake Q, the correlation between phytoplankton and zooplankton was weakened in Lake L, which meant there were still some effects of stocking *P. chinensis* on the ecological equilibrium of the plankton community, although no dramatic influences were found in Lake L yet. We also found that *P. chinensis* and Cladocera were significantly correlated, which should account for the top-down influences. Long-term successive investigations are suggested for sustainable resource utilization and maintaining biological balance.

**Keywords:** Cladocera; planktivore; zooplankton; phytoplankton

## 1. Introduction

Alien fish introduction is increasing globally, and these rising invasions have the potential to intensify future impacts [1,2]. In particular, the increase in anthropogenic activities is expected to facilitate new introductions of invasive alien fish species and subsequent invasions through pathways such as trade (e.g., aquaculture) [3,4]. However, the ecological risk of freshwater fish introduction varies by species, and the majority of freshwater fish introductions have not been identified as exerting an ecological impact, and many have great societal benefits [5]. In fact, ecological impacts of non-native fish remain largely unknown because most studies reviewed reported potential impacts rather than actual impact mechanisms, and evidence-based studies of the impacts of non-native species need to be conducted for the public perception of fish introduction risk [5,6].

In aquaculture, a manager supports naturally fluctuating stocks by stocking fish, with the assumption that the manager remembers past harvest experiences. Exploiting a stochastically fluctuating fish population facilitates the emergence of a stocking-based management panacea over time, and the social benefits of panacea formation involve

dampening natural population fluctuations and generating stability of user satisfaction [7]. Much research has sought to understand the relative strength of top-down and bottom-up control in lakes of grazing alien fishes, as sustainably managing fishery resources of human-assisted alien fishes depends on the nature of trophic control [8].

The proliferation of small-bodied fishes (e.g., obligate zooplanktivores and omnivores) in lakes often accompanied by deterioration of water quality and ecosystem function have been overlooked mainly due to their small size, shorter life spans, and lower economic value [9]. *Protosalanx chinensis* (Abbott 1901) is a small, semelparous, pelagic, annual fish that occurs in eastern Asia [10–12]. Zooplankton are the main foods of *P. chinensis* [13,14]. As a commercially important fish, *P. chinensis* has become a widely transplanted fish species in northern China, generating considerable economic benefits.

Lake L and Lake Q are shallow saline–alkali lakes located in the lower reaches of Nen River in northeastern China. They both served as green aquaculture grounds under the same management pattern with similar size, and they share the same water source. *P. chinensis* entered Lake L with the flood from Lake Dalong in 2013 and resulted in a high yield of 164t in 2016, and then the population collapsed and there was no output since 2017. Lake Q, on the other hand, has never been transplanted with *P. chinensis*. It was hypothesized that overgrazing of *P. chinensis* depleted the food resources of zooplankton and the low density of zooplankton fettered the reviving of the *P. chinensis* population. This research was designed to evaluate the subsequent effects of the *P. chinensis* population outbreak and collapse on the plankton community via comparing the two lakes.

## 2. Materials and Methods

### 2.1. Study Area and Sampling Stations

Lake L and Lake Q (46°50′4.91″ N, 124°18′1.88″ E), two adjacent lakes on the Songnen Plain, are located in western Heilongjiang Province region in northeastern China (Figure 1). They have a similar size and environmental conditions. As northern lakes, their ice-free period is from mid-April through mid-November. The long-term average water level area of Lake L and Lake Q is about 13 km$^2$ and 10 km$^2$, respectively. Lake L and Lake Q are shallow alkaline lakes with an average depth of about 2.5 m. They are under the same fishery management pattern, as they belong to the same company. The two lakes were stocked with the same species of commercial carp for fisheries. The main difference is that there was no *P. chinensis* in Lake Q. The distribution of five sampling stations were designed in Lake L (L1–L5) and Lake Q (Q1–Q5) (Figure 1).

### 2.2. Sample Collection of Plankton

Phytoplankton and zooplankton samples were collected bimonthly from each sampling site during the hatching, growing, and breeding seasons of *P. chinensis* (February to December) in 2018. Therefore, 60 phytoplankton quantitative samples and 60 zooplankton quantitative samples were obtained throughout the year. Samples (1.0 L) for phytoplankton analysis were collected in disposable plastic bottles below the surface layer of the lake (0.5 m) and fixed in Lugol's solution. Simultaneously, samples (10.0 L) for zooplankton analysis were filtered through a shallow-water plankton net (length 50 cm, diameter 20 cm, mesh size 63 μm) and fixed in a 4–5% formalin solution. After collecting the phytoplankton samples, they were transported to the laboratory and allowed to settle for 48 h. The resulting sediment was then concentrated in a single step to 60 mL before being left to settle for another 48 h. Finally, the sample was further concentrated to a volume of 30 mL for identification. The zooplankton samples were also allowed to settle for 48 h after arrival at the laboratory, and then they were concentrated to a volume of 30 mL before being identified. The identification of both the phytoplankton and zooplankton samples was conducted using a Zeiss microscope (model: primostar one). The calculation of density was converted back to the density of plankton in 1 L of water after sampling and counting by microscope. The biomass of plankton was calculated by individual average wet weight of the same genus. Processing phyto- and zooplankton, identification, enumeration, calcula-

tion of density, and biomass calculations refer to widely recognized and classic professional reference books [15–17].

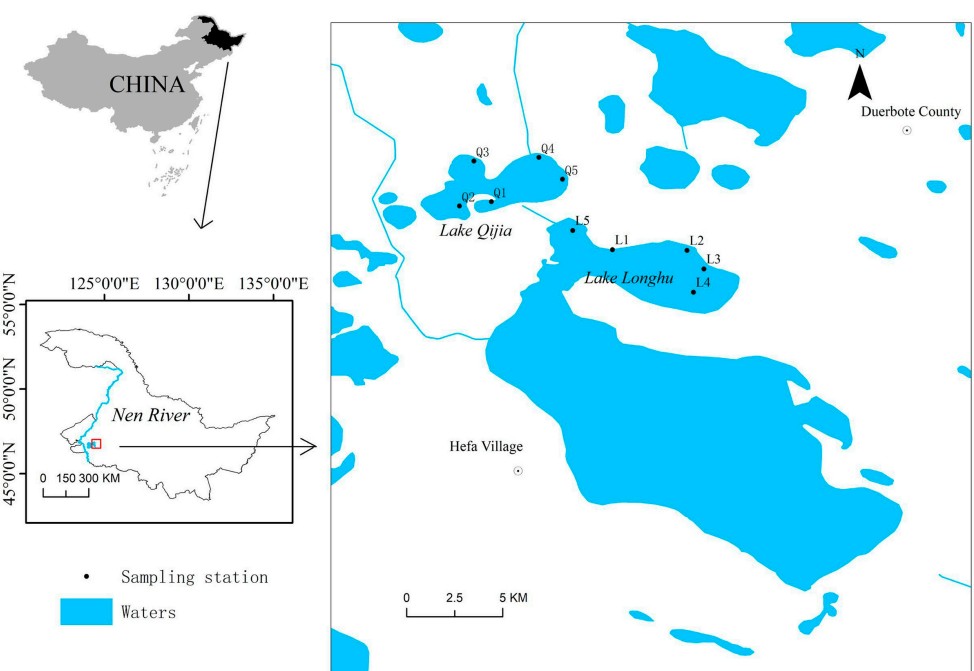

**Figure 1.** Location of sampling stations in Lake L and Lake Q.

### 2.3. Sampling and Handling Procedures of Fish

*P. chinensis* was sampled in the L1 area bimonthly from April through December 2018. Different sampling gear was used in winter and ice-free seasons to sample fish effectively. Cone trawler nets (length, 5 m; diameter, 1 m) with different mesh sizes were used for sampling fish during ice-free months. Mesh size was 1 × 1.2 mm, 2 × 1.5 mm, 3.0 × 3.0 mm, and 4.0 × 4.0 mm in April, June, August, and October, respectively. Four gill nets (length, 30 m; height, 1 m; mesh size, 10 mm, 15 mm, 20 mm, and 23 mm between opposite knots) positioned randomly were deployed in gangs, tied end to end perpendicular to the shore for 24 h per month during the freeze-up period (December). We chose 50 individuals at random for further analysis from each sample.

*P. chinensis* died as it was caught, and the standard body length was measured to the nearest 1 mm. There was no commercial exploitation in Lake L because the stock collapsed; therefore, the density of this fish did not change much through the year, and the body length and biomass increased to some extent.

### 2.4. Data Analysis

To assess phytoplankton diversity and determine the dominant species (DS), the Shannon–Wiener diversity index (*H'* [18]), Margalef diversity index (*d*) [19], Pielou index (*J*) [20], and dominance index (*Y*) [21] were calculated.

To analyze the similarity of DS in the two lakes, the Jaccard index [22] was applied to the DS of plankton to indicate the degree of similarity between lakes in different months. The Jaccard index was given by the equation:

$$C_j = j/(a + b - j) \tag{1}$$

where *a* is the number of all species occurring in community A, *b* is the number of all species occurring in community B being compared, and *j* is the number of species common to both communities.

In each survey month, independent-samples *t*-tests were performed for plankton composition, density, biomass, *H′*, *J*, and *d* in the two lakes. When the "Sig." value of "Levene's Test for Equality of Variances" is >0.05, select the "sig." value of "*t*-test for Equality of Means" in the row "Equal variances assumed". Otherwise, select the "sig." value of "*t*-test for Equality of Means" in the row "Equal variances not assumed". In order to reveal *H′*, *J*, and *d* between the two lakes, the single sample K–S (Kolmogorov–Smirnov) test was used to determine whether *H′*, *J*, and *d* had a normal distribution, and then the paired-samples Wilcoxon signed-rank test was used to compare *H′*, *J*, and *d* between the two lakes yearly. These statistical analyses were conducted using SPSS Statistics 26.0 (IBM, Armonk, NY, USA), and *p* < 0.05 was considered significant.

The statistical analyses were performed in R with the 'vegan', 'corrplot', and 'hmisc' packages. ANOSIM and principal coordinate analysis (PCoA) were analyzed by the package of 'vegan'. The R value in ANOSIM analysis is used to indicate whether there is a difference between different groups, while the *p*-value is used to indicate whether there is a significant difference. The closer R is to 1, the greater the difference between groups. Correlation analyzes were performed using the packages of 'corrplot' and 'hmisc'. The correlation analysis for density-based plankton correlations in the two lakes was calculated using Spearman's rank method. All significant differences were defined as *p* < 0.05 or *p* < 0.01.

## 3. Results

### *3.1. Plankton Community in the Two Lakes*

#### 3.1.1. Phytoplankton Composition, Density, and Biomass

A total of 133 species were recorded in two lakes: 51 Chlorophyta, 48 Bacillariophyta, 16 Cyanobacteria, 11 Euglenophyta, 2 Dinophyta, 2 Cryptophyta, 2 Xanthophyta, and 1 Chrysophyta. Chlorophyta were the most abundant phytoplankton taxa in Lake L, comprising 42.6% of total species. This was followed by Bacillariophyta (29.7%) and Cyanobacteria (13.9%). Chlorophyta comprised 33.3% of phytoplankton species in Lake Q, while Bacillariophyta was the most abundant phytoplankton taxa (38.2%). Cyanobacteria comprised 13.7% and was similar to Lake L. The mean density was $123.99 \pm 149.65 \times 10^4$ ind/L, and the total density was $991.89 \times 10^4$ ind/L in the two lakes. Chlorophyta were the most abundant taxon and contributed 46% of the total phytoplankton density, followed by Cyanobacteria (17.2%), Bacillariophyta (15.7%), Xanthophyta (12%), Cryptophyta (5.7%), Euglenophyta (3.1%), Dinophyta (0.2%), and Chrysophyta (0.1%). The mean biomass was $1.1552 \pm 1.0437$ mg/L, and the total biomass was 9.2418mg/L in the two lakes. Bacillariophyta was the highest in biomass and contributed 33.3% of the total phytoplankton biomass, followed by Cyanobacteria (21%), Euglenophyta (15.1%), Chlorophyta (13.7%), Cryptophyta (13%), Xanthophyta (2.8%), Dinophyta (1%), and Chrysophyta (0.1%).

As shown in Figure 2, the total phytoplankton composition was almost the same in the two lakes, with a difference of only 1%. Compared with Lake L, Lake Q had a slightly higher total phytoplankton density by 28%. The total phytoplankton biomass was significantly higher in Lake Q by 88% (Figure 2).

#### 3.1.2. Zooplankton Composition, Density, and Biomass

A total of 68 species were recorded in two lakes: 25 Rotifera, 16 Protozoa, 14 Copepoda, and 13 Cladocera. Rotifera was the most abundant zooplankton taxa in Lake L, comprising 29.5% of total species. This was followed by Protozoa (23.5%), Copepoda (23.5%), and Cladocera (23.5%). Protozoa comprised 26.1% of zooplankton species in Lake Q, while Rotifera was the most abundant zooplankton taxa (45.7%), Copepoda comprised 15.2%, and Cladocera comprised 13.0%. The mean density was $1303 \pm 1215.23$ ind/L, and the total density was 5212 ind/L in the two lakes. Protozoa were the most abundant taxon and contributed 59% of the total zooplankton density. This was followed by Rotifera (21.3%), Copepoda (10.4%), and Cladocera (9.3%). The average zooplankton biomass was $5.0661 \pm 4.6396$ mg/L, and the total biomass was 20.2646 mg/L in the two lakes. Cladocera

was the highest in biomass and contributed 53.1% of the total zooplankton biomass. This was followed by Copepoda (32.6%), Rotifera (13.9%), and Protozoa (0.4%).

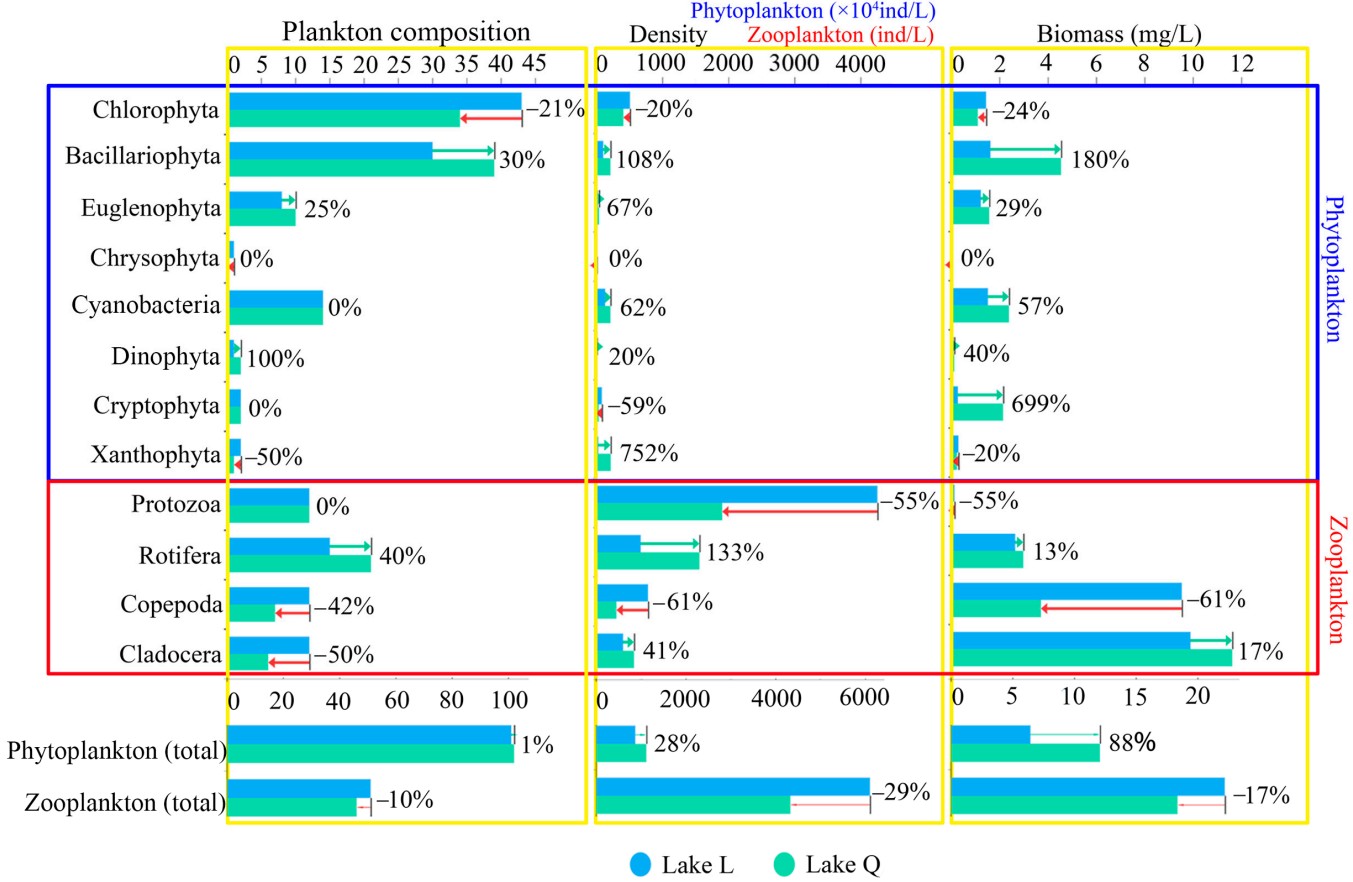

**Figure 2.** Plankton composition, density, and biomass in the two lakes.

Compared with Lake Q, the total zooplankton composition was higher in Lake L by 10%, with Copepoda and Cladocera contributing the most. Lake L had a higher total zooplankton density by 29%, with Protozoa and Copepoda contributing the most. The total zooplankton biomass was higher in Lake L by 17%, with Copepoda contributing the most to the higher biomass (Figure 2).

In each survey month, the composition, density, and biomass of plankton in Lake L and Q were compared using independent-samples *t*-tests to investigate significant differences. The results showed that significant differences existed in phytoplankton composition in the two lakes in October; in phytoplankton density in the two lakes in April, October, and December; and in phytoplankton biomass in August and October. The zooplankton composition differed significantly in the two lakes in April, August, October, and December, while zooplankton density exhibited significant differences in April, October, and December. Furthermore, zooplankton biomass differed significantly in April and August between the two lakes (Table 1).

**Table 1.** Independent-samples *t*-test of plankton composition, density, and biomass.

| Item | Month | Phytoplankton Mean ± Std. Deviation Lake L | Lake Q | Sig. | Zooplankton Mean ± Std. Deviation Lake L | Lake Q | Sig. |
|------|-------|---------------------------------------------|--------|------|-------------------------------------------|--------|------|
| Composition | Feb. | 19.6 ± 6.269 | 19.4 ± 5.32 | 0.958 | 11.4 ± 2.51 | 12.4 ± 2.074 | 0.512 |
| | Apr. | 8.6 ± 0.894 | 8.6 ± 3.647 | 1 | 10.6 ± 1.14 | 13.2 ± 1.789 | 0.025 * |
| | Jun. | 8.4 ± 1.817 | 9 ± 3.808 | 0.759 | 12.6 ± 2.302 | 10.4 ± 1.14 | 0.092 |
| | Aug. | 24.8 ± 6.907 | 29.8 ± 8.167 | 0.326 | 20.6 ± 0.894 | 14.6 ± 1.817 | 0.001 * |
| | Oct. | 19 ± 2.55 | 26.4 ± 2.966 | 0.003 * | 11.4 ± 2.302 | 4.8 ± 1.304 | 0.001 * |
| | Dec. | 15.8 ± 9.176 | 9.8 ± 3.633 | 0.23 | 8 ± 3.536 | 3.4 ± 1.14 | 0.024 * |
| Density | Feb. | 209.33 ± 65.0327 | 235.44 ± 93.822 | 0.623 | 2285.4 ± 1670.583 | 1042.2 ± 474.183 | 0.175 |
| | Apr. | 114.46 ± 15.226 | 65.26 ± 43.763 | 0.045 * | 483.6 ± 184.621 | 1046.4 ± 369.307 | 0.016 * |
| | Jun. | 70.28 ± 12.424 | 82.83 ± 47.292 | 0.593 | 773.4 ± 330.132 | 1512.6 ± 651.93 | 0.054 |
| | Aug. | 344.87 ± 210.59 | 520.07 ± 213.405 | 0.228 | 1353.6 ± 1211.018 | 1331.4 ± 938.673 | 0.975 |
| | Oct. | 160.14 ± 27.973 | 306.72 ± 59.646 | 0.001 * | 1806 ± 1077.785 | 177.6 ± 147.861 | 0.027 * |
| | Dec. | 201.3 ± 43.993 | 148.09 ± 25.225 | 0.047 * | 616.2 ± 397.727 | 80.4 ± 144.63 | 0.022 * |
| Biomass | Feb. | 1.7192 ± 0.875 | 2.6403 ± 1.427 | 0.253 | 2.3445 ± 1.49 | 0.7977 ± 0.424 | 0.08 |
| | Apr. | 0.4977 ± 0.175 | 0.4293 ± 0.339 | 0.699 | 3.9683 ± 0.802 | 2.5178 ± 1.008 | 0.036 * |
| | Jun. | 0.2405 ± 0.109 | 0.5084 ± 0.379 | 0.193 | 10.6277 ± 5.563 | 14.4863 ± 8.003 | 0.402 |
| | Aug. | 2.8504 ± 1.849 | 6.9566 ± 2.514 | 0.019 * | 7.7738 ± 1.102 | 3.0207 ± 1.194 | 0.0002 * |
| | Oct. | 1.8548 ± 0.862 | 3.5296 ± 0.772 | 0.012 * | 0.7236 ± 0.432 | 0.9079 ± 0.334 | 0.472 |
| | Dec. | 0.5706 ± 0.439 | 0.4042 ± 0.227 | 0.473 | 1.1819 ± 1.139 | 0.2846 ± 0.206 | 0.121 |

Notes: The unit of measurement for the phytoplankton density was $10^4$ ind/L, and the unit for the zooplankton density was ind/L. The asterisk (*) indicates that the associated significance level was <0.05.

### 3.1.3. Dominant Species (DS)

The DS are shown in Table 2. Phytoplankton was dominated by Chlorophyta (15) and Cyanobacteria (9), followed by Bacillariophyta (7), Euglenophyta (2), Cryptophyta (2), and Xanthophyta (1). Zooplankton was dominated by Rotifera (13) and Protozoa (13), followed by Cladocera (7) and Copepoda (6) (Figure 3).

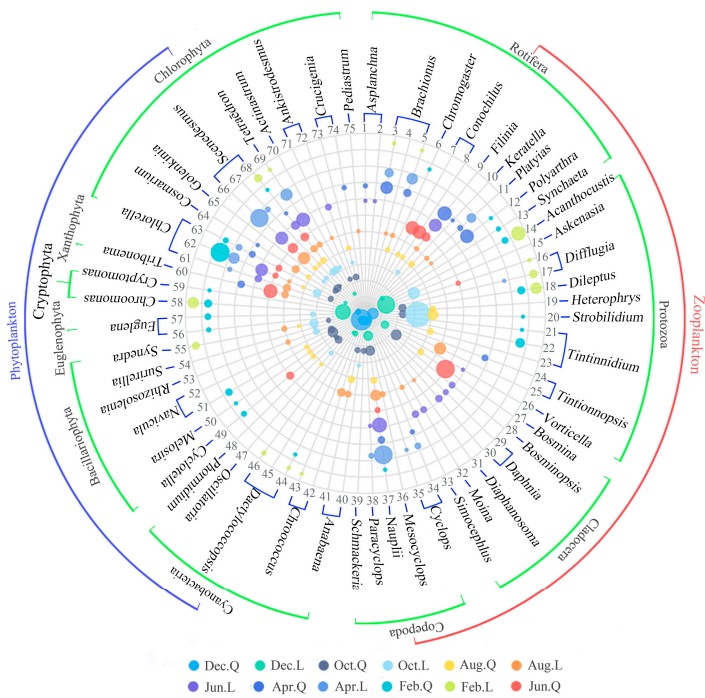

**Figure 3.** Plankton DS and dominance in the two lakes. The diameter of the circle in the figure represents the degree of dominance of the DS. Feb., Apr., Jun., Aug., Oct., and Dec. were abbreviations for the months of February, April, June, August, October, and December, respectively. L and Q represent Lake L and Lake Q, respectively. See Table 2 for the numbers representing the DS.

**Table 2.** Dominant plankton species list.

| No. | Dominant Species | No. | Dominant Species | No. | Dominant Species |
| --- | --- | --- | --- | --- | --- |
| | Zooplankton | 26 | *Vorticella* sp. | 51 | *Navicula rhynchocephala* |
| 1 | *Asplanchna brightwelli* | 27 | *Bosmina longirostris* | 52 | *Navicula* sp. |
| 2 | *Asplanchna priodonta* | 28 | *Bosminopsis deitersi* | 53 | *Rhizosolenia longiseta* |
| 3 | *Brachionus angularis* | 29 | *Daphnia cristata* | 54 | *Surirellia ovate* var. *pinnata* |
| 4 | *Brachionus calyciflorus* | 30 | *Daphnia pulex* | 55 | *Synedra acus* |
| 5 | *Brachionus urceus* | 31 | *Diaphanosoma branchyurum* | 56 | *Euglena acus* |
| 6 | *Chromogaster ovalis* | 32 | *Moina micrur* | 57 | *Euglena geniculate* |
| 7 | *Conochilus dossnarius* | 33 | *Simocephlus vetulus* | 58 | *Chroomonas acuta* |
| 8 | *Conochilus unicornis* | 34 | *Cyclops strenuus* | 59 | *Cryptomonas erosa* |
| 9 | *Filinia longiseta* | 35 | *Cyclops vicinus* | 60 | *Tribonema minus* |
| 10 | *Keratella quadrata* | 36 | *Mesocyclops leuckarti* | 61 | *Chlorella ellipsoidea* |
| 11 | *Platyias quadricornis* | 37 | *Nauplii* | 62 | *Chlorella pyrenoidosa* |
| 12 | *Polyarthra trigla* | 38 | *Paracyclops affinis* | 63 | *Chlorella vulgaris* |
| 13 | *Synchaeta longipes* | 39 | *Schmackeria inopinus* | 64 | *Cosmarium nastutum* |
| 14 | *Acanthocustis* sp. | | Phytoplankton | 65 | *Golenkinia radiate* |
| 15 | *Askenasia* sp. | 40 | *Anabaena circinalis* | 66 | *Scenedesmus bijuga* |
| 16 | *Difflugia globulosa* | 41 | *Anabaena oscillarioides* | 67 | *Scenedesmus obliquus* |
| 17 | *Difflugia oblonga* | 42 | *Chroococcus minor* | 68 | *Scenedesmus quadricauda* |
| 18 | *Dileptus* sp. | 43 | *Chroococcus tenax* | 69 | *Tetraēdron tumidulum* |
| 19 | *Heterophrys* sp. | 44 | *Dactylococcopsis irregularis* | 70 | *Actinastrum hantzschii* |
| 20 | *Strobilidium* sp. | 45 | *Dactylococcopsis rhaphidioides* | 71 | *Ankistrodesmus falcatus* |
| 21 | *Tintinnidium entzii* | 46 | *Dactylococcopsis rhaphidioides* f. *falciformis* | 72 | *Ankistrodesmus falcatus* var. *mirabilis* |
| 22 | *Tintinnidium fluviatile* | 47 | *Oscillatoria* sp. | 73 | *Crucigenia tetrapedia* |
| 23 | *Tintinnidium pusillum* | 48 | *Phormidium tenue* | 74 | *Crucigenia puadrata* |
| 24 | *Tintionnopsis sinensis* | 49 | *Cyclotella comta* | 75 | *Pediastrum boryanum* |
| 25 | *Tintionnopsis wangi* | 50 | *Melosira granulate* var. *angustissima* | | |

In Lake L, DS of Rotifera occurred from February to December, and the total dominance (TD) was relatively high in April (0.33), June (0.22), and August (0.13) and relatively low in February (0.04), October (0.03), and December (0.03). DS of Protozoa also occurred from February to December, and the DS was relatively high in October (0.86), February (0.46), and December (0.43), and it was relatively low in August (0.16), April (0.02), and June (0.02). DS of Copepoda occurred from April to August and December, and the TD decreased gradually in April (0.49), June (0.26), August (0.17), and December (0.11). DS of Cladocera occurred in June and August, and the TD in June (0.26) was higher than that in August (0.05). In Lake Q, DS of Protozoa occurred from February to December, and the TD was relatively high in February (0.26), August (0.23), October (0.14), and December (0.15) and relatively low in April (0.05) and June (0.03). DS of Copepoda also occurred from February to December, and the TD was relatively high in October (0.15), April (0.09), December (0.09), August (0.05), and June (0.04) and relatively low in February (0.02). DS of Rotifera occurred from February to August, and the TD was relatively high in April (0.75) and June (0.49), and it was relatively low in August (0.14) and February (0.07). DS of Cladocera occurred from June to December, and the TD was relatively high in June (0.37) and relatively low in October (0.13), August (0.05), and December (0.05) (Figure 3).

In Lake L, DS of Chlorophyta occurred from February to December, and the TD of was relatively high in April (0.78), June (0.76), August (0.46), and October (0.3) and relatively low in February (0.09) and December (0.04). DS of Cyanobacteria occurred in February and from August to December, and Bacillariophyta occurred in February, August, and October. Cryptophyta occurred in February and December, Euglenophyta occurred in October, and Xanthophyta occurred in August. In Lake Q, DS of Chlorophyta occurred from February to December, and the TD was relatively high in February (0.45), June (0.28), and October (0.25), and it was relatively low in April (0.12), August (0.18), and December (0.02). DS of

Bacillariophyta occurred in February and from June to October, and the TD was high in October (0.24). DS of Cyanobacteria occurred in February and from August to December, and the highest TD was in December (0.68). DS of Xanthophyta occurred in April, June, and December, and the highest total dominance was in June (0.21). DS of Cryptophyta occurred in February and December, Euglenophyta occurred in February, and neither of the TD values were high (Figure 3).

The *Cj* values of phytoplankton and zooplankton between the two lakes were not high (index values below 0.50 indicate low similarity; values above 0.50 indicate high similarity), with annual averages of 0.26 and 0.18, respectively. Phytoplankton had the highest similarity in December with a *Cj* of 0.38 and the lowest similarity in April with a *Cj* of 0.1. The *Cj* was 0.2, 0.33, 0.32, and 0.25 in February, June, August, and October, respectively. For zooplankton, the similarity was the highest in April with a *Cj* of 0.38, and the similarity was the lowest in August with *Cj* of 0.12. The *Cj* was 0.15, 0.13, 0.14, and 0.14 in February, June, October, and December, respectively.

In Lake L, Protozoa and Rotifera were dominant in every survey month, especially when the total dominance of the dominant species of Protozoa suddenly rose to 0.86. Cladocera dominated in June and August, and the total dominance of dominant species decreased. Copepoda dominated from April to August, and the total dominance of the dominant species gradually decreased. After not being dominant in October, Copepoda dominated again in December (Table 3). In Lake Q, Rotifera were dominant from February to August, and Protozoa were dominant in every survey month. Compared with L, the total dominance of the dominant species of Protozoa did not increase or decrease significantly. In Lake Q, Cladocera dominated from June to December. Copepoda were dominant in every survey month, and the total dominance of dominant species did not increase or decrease significantly (Table 3).

**Table 3.** Numbers and total dominance of zooplankton DS in the survey months.

| Month/Lake | Rotifera | | Protozoa | | Cladocera | | Copepoda | |
|---|---|---|---|---|---|---|---|---|
| | NDS | TDS | NDS | TDS | NDS | TDS | NDS | TDS |
| Feb./L | 2 | 0.04 | 4 | 0.46 | 0 | 0 | 0 | 0 |
| Apr./L | 3 | 0.33 | 1 | 0.02 | 0 | 0 | 4 | 0.49 |
| Jun./L | 3 | 0.22 | 1 | 0.02 | 5 | 0.26 | 2 | 0.26 |
| Aug./L | 5 | 0.13 | 1 | 0.16 | 2 | 0.05 | 2 | 0.17 |
| Oct./L | 1 | 0.03 | 2 | 0.86 | 0 | 0 | 0 | 0 |
| Dec./L | 1 | 0.03 | 2 | 0.43 | 0 | 0 | 1 | 0.11 |
| Feb./Q | 3 | 0.07 | 5 | 0.26 | 0 | 0 | 1 | 0.02 |
| Apr./Q | 7 | 0.75 | 1 | 0.05 | 0 | 0 | 2 | 0.09 |
| Jun./Q | 3 | 0.49 | 1 | 0.03 | 1 | 0.37 | 1 | 0.04 |
| Aug./Q | 4 | 0.14 | 3 | 0.23 | 1 | 0.05 | 1 | 0.05 |
| Oct./Q | 0 | 0 | 2 | 0.14 | 1 | 0.13 | 2 | 0.15 |
| Dec./Q | 0 | 0 | 1 | 0.15 | 2 | 0.05 | 1 | 0.09 |

Note: NDS: numbers of DS. TDS: total dominance of DS in the survey month.

In Lake L, Chlorophyta dominated in every survey month, and the total dominance of dominant species was higher from April to August and gradually decreased. Cyanobacteria dominated in February and August–December. Bacillariophyta dominated in February and August–October. Other phytoplankton had a certain degree of dominance in different months (Table 4). In Lake Q, Chlorophyta dominated in every survey month, and the total dominance of dominant species was the highest in February, and there was no obvious increase or decrease in other months. Cyanobacteria dominated in February and August–December, and the total dominance of dominant species was higher in December. Bacillariophyta dominated in February and June–October. Other phytoplankton had a certain degree of dominance in different months (Table 4).

**Table 4.** Numbers and total dominance of phytoplankton DS in the survey months.

| Month/Lake | Cyanobacteria | | Bacillariophyta | | Euglenophyta | | Cryptophyta | | Xanthophyta | | Chlorophyta | |
|---|---|---|---|---|---|---|---|---|---|---|---|---|
| | NDS | TDS | NDS | TDS | NDS | TDS | NDS | TDS | NDS | TDS | NDS | TDS |
| Feb./L | 3 | 0.07 | 1 | 0.07 | 0 | 0 | 1 | 0.14 | 0 | 0 | 2 | 0.09 |
| Apr./L | 0 | 0 | 0 | 0 | 0 | 0 | 0 | 0 | 0 | 0 | 6 | 0.78 |
| Jun./L | 0 | 0 | 0 | 0 | 0 | 0 | 0 | 0 | 0 | 0 | 6 | 0.76 |
| Aug./L | 2 | 0.13 | 2 | 0.07 | 0 | 0 | 0 | 0 | 1 | 0.06 | 8 | 0.46 |
| Oct./L | 2 | 0.05 | 3 | 0.16 | 1 | 0.04 | 0 | 0 | 0 | 0 | 7 | 0.3 |
| Dec./L | 2 | 0.06 | 0 | 0 | 0 | 0 | 1 | 0.27 | 0 | 0 | 2 | 0.04 |
| Feb./Q | 1 | 0.02 | 3 | 0.12 | 2 | 0.08 | 2 | 0.09 | 0 | 0 | 3 | 0.45 |
| Apr./Q | 0 | 0 | 0 | 0 | 0 | 0 | 0 | 0 | 1 | 0.06 | 4 | 0.12 |
| Jun./Q | 0 | 0 | 1 | 0.07 | 0 | 0 | 0 | 0 | 1 | 0.21 | 4 | 0.28 |
| Aug./Q | 2 | 0.08 | 4 | 0.11 | 0 | 0 | 0 | 0 | 0 | 0 | 6 | 0.18 |
| Oct./Q | 2 | 0.11 | 4 | 0.24 | 0 | 0 | 0 | 0 | 0 | 0 | 6 | 0.25 |
| Dec./Q | 3 | 0.68 | 0 | 0 | 0 | 0 | 1 | 0.02 | 1 | 0.04 | 1 | 0.02 |

Note: NDS: numbers of DS. TDS: total dominance of DS in the survey month.

3.1.4. Plankton Diversity in the Two Lakes

The complete 6-month data for plankton diversity in the two lakes are shown in Figure 4. Over the course of 6 months, the total phytoplankton diversity index value in the two lakes (Feb.: 24.22, Apr.: 14.53, Jun.: 15.02, Aug.: 29.16, Oct.: 28.32, Dec.: 16.85, and total: 128.1) was higher than that of zooplankton (Feb.: 11.93, Apr.: 14.5, Jun.: 13.45, Aug.: 18.38, Oct.: 8.72, Dec.: 8.03, and total:75.01). Plankton diversity in August was higher than other months (Figure 4), which was caused by both phytoplankton and zooplankton. Diversity in Lake L slightly outnumbered Lake Q regardless of phytoplankton (Lake L: 64.14 and Lake Q: 63.96) and zooplankton (Lake L: 39.29 and Lake Q: 35.72) (Figure 4).

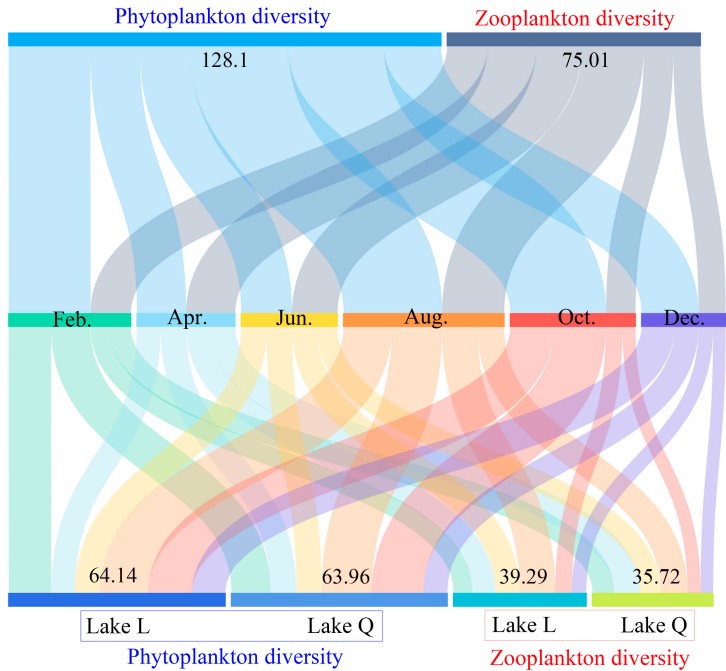

**Figure 4.** Plankton diversity in the two lakes. The length of the color-coded bars represented the magnitude of the value. This figure was divided into three layers. The top layer shows the sum of three diversity indices ($H'$, $J$, and $d$) for phytoplankton and zooplankton. The bottom layer shows the sum of the three diversity index values for plankton in the lakes L and Q, respectively. The middle layer shows the monthly variation in the sum of the three diversity index values.

Table 5 shows the significance analysis of the three diversity indices of plankton in the two lakes in each survey month based on the independent-samples *t*-test. The *H'* and *J* of phytoplankton were significantly different in October. The J of phytoplankton in August was significantly different. Zooplankton H' was significantly different in April. J was significantly different in April and December. d was significantly different from June to October (Table 5).

**Table 5.** Independent-samples *t*-test of *H'*, *J*, and *d* in the survey months.

| Index | Month | Phytoplankton | | | Zooplankton | | |
|---|---|---|---|---|---|---|---|
| | | Mean ± Std. Deviation | | Sig. | Mean ± Std. Deviation | | Sig. |
| | | Lake L | Lake Q | | Lake L | Lake Q | |
| *H'* | Feb. | 3.668 ± 0.486 | 3.172 ± 0.537 | 0.164 | 1.782 ± 0.644 | 1.986 ± 0.312 | 0.541 |
| | Apr. | 2.504 ± 0.269 | 2.472 ± 0.56 | 0.912 | 2.432 ± 0.1 | 2.884 ± 0.237 | 0.004 * |
| | Jun. | 2.648 ± 0.343 | 2.476 ± 0.425 | 0.502 | 2.536 ± 0.547 | 2.344 ± 0.313 | 0.515 |
| | Aug. | 3.934 ± 0.495 | 3.508 ± 0.507 | 0.216 | 3.194 ± 0.599 | 2.518 ± 0.559 | 0.102 |
| | Oct. | 3.898 ± 0.148 | 4.218 ± 0.203 | 0.022 * | 1.274 ± 0.36 | 1.36 ± 0.656 | 0.804 |
| | Dec. | 2.722 ± 1.239 | 2.308 ± 0.461 | 0.515 | 1.306 ± 0.707 | 1.27 ± 0.49 | 0.928 |
| *J* | Feb. | 0.864 ± 0.047 | 0.75 ± 0.126 | 0.094 | 0.502 ± 0.153 | 0.548 ± 0.07 | 0.564 |
| | Apr. | 0.81 ± 0.093 | 0.824 ± 0.061 | 0.786 | 0.716 ± 0.032 | 0.778 ± 0.036 | 0.021 * |
| | Jun. | 0.868 ± 0.029 | 0.824 ± 0.082 | 0.29 | 0.694 ± 0.117 | 0.664 ± 0.068 | 0.633 |
| | Aug. | 0.858 ± 0.044 | 0.722 ± 0.055 | 0.003 * | 0.732 ± 0.142 | 0.638 ± 0.135 | 0.316 |
| | Oct. | 0.92 ± 0.014 | 0.894 ± 0.015 | 0.023 * | 0.364 ± 0.099 | 0.606 ± 0.248 | 0.096 |
| | Dec. | 0.69 ± 0.175 | 0.728 ± 0.018 | 0.654 | 0.43 ± 0.152 | 0.792 ± 0.279 | 0.035 * |
| *d* | Feb. | 7.996 ± 2.301 | 7.77 ± 1.744 | 0.865 | 3.276 ± 0.975 | 3.834 ± 0.622 | 0.312 |
| | Apr. | 3.698 ± 0.406 | 4.23 ± 1.54 | 0.492 | 3.602 ± 0.281 | 4.088 ± 0.744 | 0.209 |
| | Jun. | 3.998 ± 0.856 | 4.2 ± 1.54 | 0.804 | 4.048 ± 0.687 | 3.168 ± 0.363 | 0.035 * |
| | Aug. | 9.516 ± 2.085 | 10.618 ± 2.299 | 0.45 | 6.562 ± 0.673 | 4.742 ± 0.83 | 0.005 * |
| | Oct. | 8.174 ± 1.044 | 10.222 ± 0.874 | 0.01 * | 3.256 ± 0.633 | 1.862 ± 0.718 | 0.012 * |
| | Dec. | 6.37 ± 3.718 | 4.028 ± 1.601 | 0.232 | 2.592 ± 1.265 | 1.638 ± 0.434 | 0.149 |

Note: The asterisk (*) indicates that the associated significance level was <0.05.

Paired-samples Wilcoxon signed-rank tests to *H'*, *J*, and *d* between lakes were summarized in Table 6. The results showed that most of the plankton diversity indices were not significantly different between the two lakes, except that phytoplankton *J* and zooplankton *d* were relatively higher in lake L (Table 6).

**Table 6.** Differences of *H'*, *J*, and *d* between the two lakes yearly.

| Diversity Indices | | Z Value | *p* Value | Mean Value | |
|---|---|---|---|---|---|
| | | | | Lake L | Lake Q |
| *H'* | Phytoplankton | −1.399 | >0.05 | 3.229 ± 0.8348 | 3.026 ± 0.8132 |
| | Zooplankton | −0.257 | >0.05 | 2.087 ± 0.8590 | 2.060 ± 0.7283 |
| *J* | Phytoplankton | −1.982 | <0.05 * | 0.835 ± 0.1075 | 0.790 ± 0.0901 |
| | Zooplankton | 1.72 | >0.05 | 0.573 ± 0.1869 | 0.671 ± 0.1768 |
| *d* | Phytoplankton | 1.224 | >0.05 | 6.625 ± 2.8928 | 6.845 ± 3.2532 |
| | Zooplankton | −2.551 | <0.05 * | 3.889 ± 1.4899 | 3.222 ± 1.2978 |

Note: The asterisk (*) indicates that the associated significance level was <0.05.

### 3.2. Plankton Community Similarity between the Two Lakes

A similarity analysis of plankton density obtained in the principal coordinate analysis (PCoA) and the experimental parameters showed that lake- and month-related differences aligned along coordinate 2 in phytoplankton and zooplankton, explaining 16.62% and 16.12% of the total variation, respectively (Figure 5a,b). The two scatterplots generated by PCoA analysis showed that the phytoplankton in Lake L clustered in one area in April

and June, while they were more scattered in other months (Figure 5a). In contrast, the phytoplankton in Lake Q were more scattered overall and did not exhibit similar clustering patterns as those in Lake L in certain months (Figure 5a). For zooplankton, those in Lake L clustered in one area in October, while they were relatively scattered in other months (Figure 5b). The zooplankton in Lake Q were also relatively scattered, which contrasted with the clustering pattern of those in Lake L in October (Figure 5b).

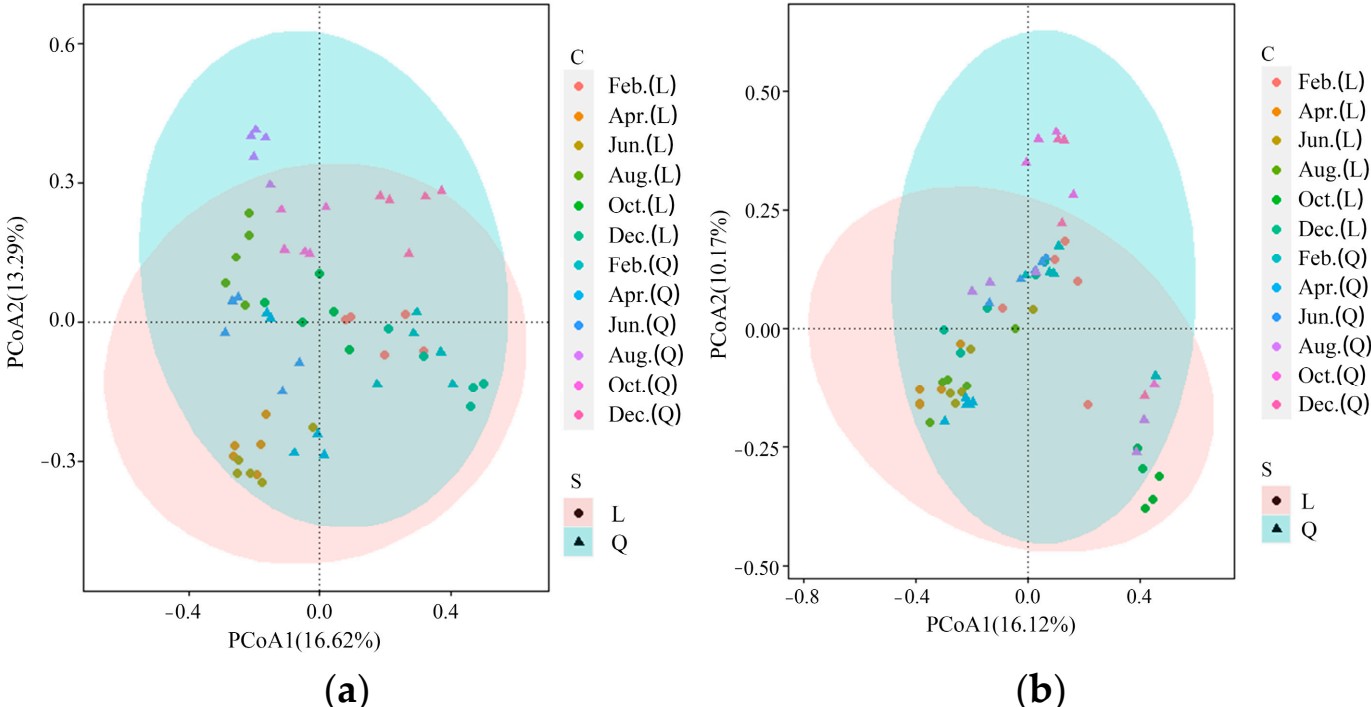

**Figure 5.** Principal coordinate analysis (PCoA) of phytoplankton density (**a**) and zooplankton density (**b**) based on the Bray–Curtis dissimilarity at the six sampled time points in two lakes. L: Lake L, Q: Lake Q.

Plankton density differences between lakes were tested with the use of one-way ANOSIM, and the significance was computed by permutation of group membership set to 9999 replicates. The statistical testing led to its statistical confirmation at the level of 5% (Table 7). According to ANOSIM analysis, the differences between the plankton of the two lakes in most months were significant or extremely significant, except for zooplankton with $p$-values greater than 0.05 in February (Table 7). Specifically, the differences in phytoplankton between the two lakes were greater in December and October, while the greatest difference in zooplankton was observed in April (Table 7). Additionally, it was noted that the R value for zooplankton was relatively high when the R value for phytoplankton was relatively low, suggesting some correlation or association between the two lakes.

**Table 7.** ANOSIM randomization test to confirm statistically significant differences between lakes of months.

| Pairwise Comparisons between Lakes of Months | Zooplankton | | Phytoplankton | |
|---|---|---|---|---|
| | R Value | $p$ Value | R Value | $p$ Value |
| February | 0.072 | 0.317 | 0.608 | 0.006 |
| April | 0.964 | 0.006 | 0.524 | 0.011 |
| June | 0.872 | 0.012 | 0.384 | 0.008 |
| August | 0.72 | 0.01 | 0.872 | 0.008 |
| October | 0.82 | 0.009 | 0.948 | 0.015 |
| December | 0.74 | 0.009 | 1 | 0.013 |

### 3.3. Correlations among Phytoplankton, Zooplankton, and P. chinensis

The correlations among categories of phytoplankton and zooplankton are shown in Figure 6. The proportion of significant correlations among categories of phytoplankton and zooplankton was higher in Lake Q (32.1%) than in Lake L (10.7%). Furthermore, 17.9% of the correlations among categories of phytoplankton and zooplankton were negatively significant in Lake Q, and the corresponding proportion was 7.1% in Lake L. Additionally, 14.3% of the correlations among the categories of phytoplankton and zooplankton were positively significant in Lake Q, and the corresponding proportion was 3.6% in Lake L.

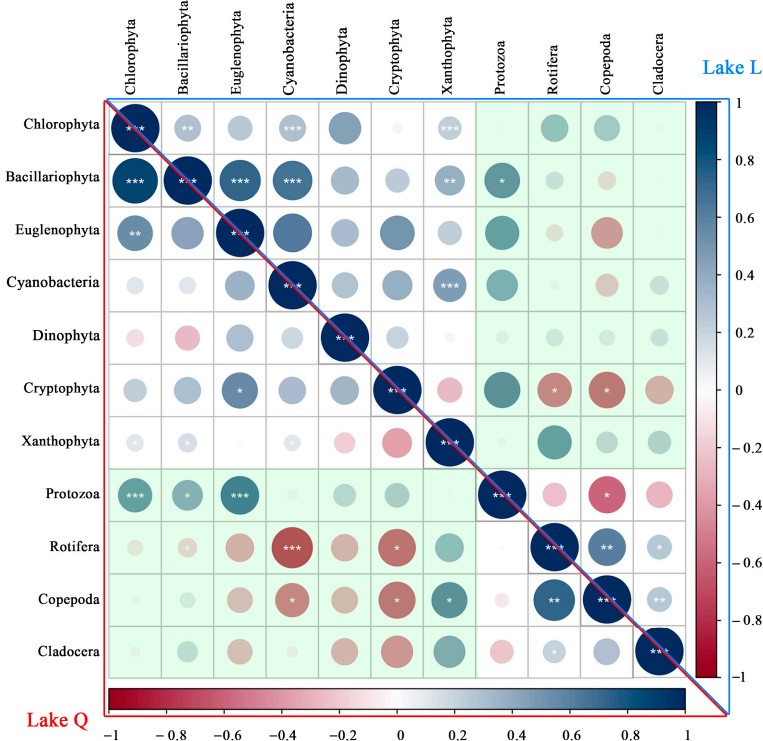

**Figure 6.** Density-based plankton correlations in the two lakes. The lower triangle represents the correlation of plankton in Lake Q, and the upper triangle represents the correlation of plankton in Lake L. The green area emphasized the correlation between phytoplankton and zooplankton. * *p* < 0.05, ** *p* < 0.01, *** *p* < 0.001.

The categories of phytoplankton showed significant positive correlations only, and the significant proportion in Lake Q (28.6%) was slightly lower than in Lake L (33.3%). Moreover, the proportion of significant correlations among categories of zooplankton were also lower in Lake Q (33.3%) than in Lake L (66.7%). Furthermore, 50.0% of the correlations among categories of zooplankton were positively significant in Lake L, and the corresponding proportion was 33.3% in Lake Q. Additionally, 16.7% of the correlations among the categories of zooplankton were negatively significant in Lake L, while no negative correlation was found in Lake Q. These results suggested that there were differences in the correlation of plankton between the two lakes and that the introduction of fish might have triggered changes in the correlation of plankton.

The correlations between *P. chinensis* body length and the densities of phytoplankton categories were all not significant. In other words, it could not be confirmed that there was any significant relationship between *P. chinensis* growth and phytoplankton density (Figure 7). However, the results indicated that there was a significant positive correlation between *P. chinensis* body length and the density of Cladocera, suggesting *P. chinensis* grazing could be a factor affecting the zooplankton community in Lake L.

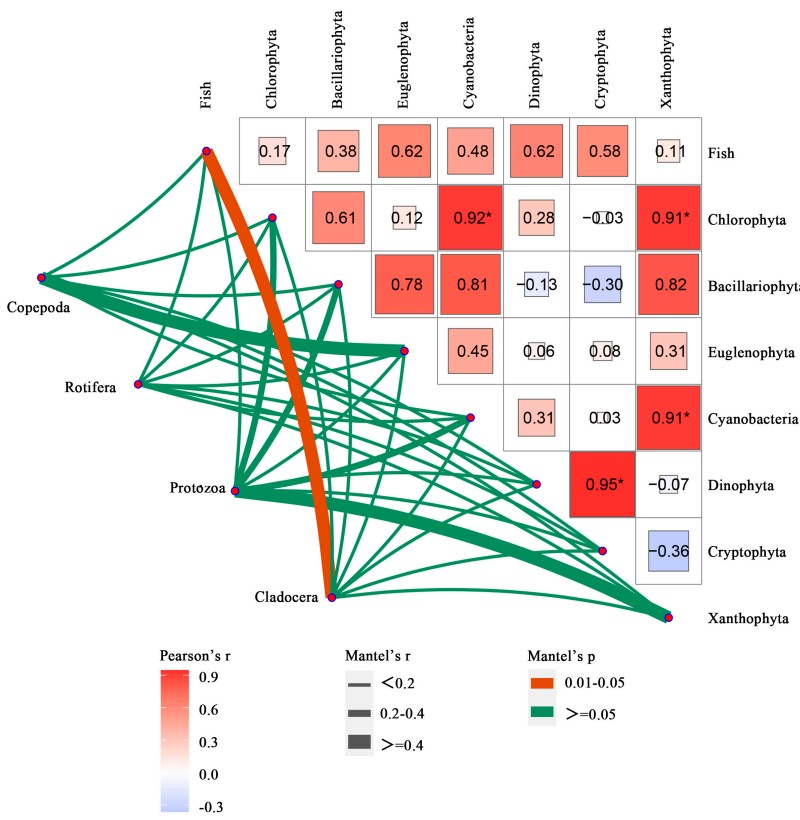

**Figure 7.** The correlations of zooplankton density with phytoplankton density and body length of *P. chinensis*. Significance: * *p* < 0.05.

## 4. Discussion

### 4.1. Zooplankton Resources Not Limitting P. chinensis Revival in Lake L

Zooplanktivorous fish mainly feed on small zooplankton at the larvae and early juvenile stage, such as Rotifera, Cyclopoida, and *Bosmina*, and then they gradually transition to feeding on large zooplankton such as Cladocera [23]. Previous studies on the likely effects of stocking planktivorous fish presented an overall ambiguous picture [24]. Some researchers suggested that stocked planktivorous fish could help stabilize phytoplankton density and diversity in shallow lakes by filtering phytoplankton, while others held the complete opposite opinion that fish predation reduced zooplankton and boosted small-sized algae, which would affect the plankton community stability of lakes [25–29]. This study revealed that the zooplankton density and biomass were not less in Lake L (*P. chinensis* stocked) than in Lake Q (*P. chinensis* free) in each month (Table 1, Figure 2). Based on phytoplankton or zooplankton, the biodiversity of Lake L was slightly higher than that of Lake Q (Figure 4), and there was no significant difference in the phytoplankton biodiversity of Lake L and Lake Q in most months (Table 5). Hence, the zooplankton resources were still abundant and the community structure was also stable after the *P. chinensis* population collapse, and zooplankton resources should not be a limiting factor of *P. chinensis* population recovery in Lake L. Overall, the outbreak of the *P. chinensis* population either did not deplete the zooplankton in Lake L or the zooplankton have strong resilience after being overgrazed.

Therefore, the zooplankton community should be investigated during the explosion and collapse of the *P. chinensis* population to determine whether *P. chinensis* had a significant effect on the zooplankton resources and to understand the relationship between the dynamics of zooplankton resources and the population of *P. chinensis*.

### 4.2. Influnces of Stocking P. chinensis on the Plankton Community

The phytoplankton and zooplankton communities are crucial components of freshwater ecosystems [30], playing essential roles in maintaining biological balance in aquatic environments [31]. Several studies have demonstrated that planktivorous fish predation caused a decrease in abundance of big Cladocera (e.g *Daphnia*) while favoring small Cladocera (e.g *Bosmina* and *Chydorus*), Copepoda, and Rotifera [32–34]. Short-term planktivorous fish predation could result in the destruction of the resilience of zooplankton communities and hinder the recovery of phytoplankton controlled by zooplankton grazing [35]. Fish predation could also have indirect effects on phytoplankton communities through trophic cascades [36–38] or nutrient recycling [39–41]. Specifically, Cladocera were the most important planktonic herbivores in freshwater lakes and could consume as much as 80–100% of the phytoplankton biomass. Higher predation risk, mostly from fish, was the key factor for the dominance of particularly small-sized zooplankton. Fish had an indirect effect on the lower trophic levels by reducing the overall zooplankton biomass and grazing pressure on the phytoplankton community. Fish could increase nutrient loading to phytoplankton in several ways, including through excreta and feces, increased nutrient excretion by zooplankton, and fish mortality and decay of fish carcasses [35]. This implied a positive or negative correlation between phytoplankton and zooplankton.

In this research, the correlation among phytoplankton and zooplankton was weaker in Lake L than in Lake Q, which should be a result of top-down influences. *P. chinensis* mainly preys on Copepoda and Cladocera [14,42]. The top-down effect was stronger than the bottom-up effect in October and December, possibly due to the low temperature. In October and December, the zooplankton composition and density were significantly different in Lake L compared to Lake Q. The small zooplankton of Rotifera and Protozoa became predominant in Lake L, while both Copepoda and Cladocera were not predominant. In particular, the reduction in big zooplankton predominance caused the predominance of protozoa to increase drastically from 0.16 in August to 0.86 in October. The top-down effect also caused significant differences in phytoplankton resources, including composition, density, biomass, and the three diversity indices, between the two lakes in October. However, these effects were not sustained over time, and the differences vanished in December. As the large zooplankton decreased in dominance, the total dominance of Protozoa and Rotifera suddenly increased in October and December, which was very different from the dynamics in Lake Q (*P. chinensis* free). The higher declining rate of Cladocera density and biomass from June to August was attributed to *P. chinensis* transitioning to feed on Cladocera in June [42].

### 4.3. Sustainable Fishery Management

As we have previously discussed, the low population density of *P. chinensis* could not be attributed to the insufficiency of zooplankton resources, and we need to identify critical factors that could enhance the population and yield. It has been reported that the high population density of *P. chinensis* led to delayed female maturation in the winter of 2016, with limited availability of males for mating leading to inefficient natural reproduction [43]. The delay in female maturation could be attributed to the inadequacy of food resources. However, the current research indicated that the zooplankton density could recover quickly in Lake L. Therefore, to develop the population and recover the yield of *P. chinensis* in Lake L, we need to focus on the bottleneck of *P. chinensis* reproduction and recruitment. Although the effects of *P. chinensis* stocking on the ecological equilibrium of the plankton community in Lake L were detected, long-term and successive investigations are necessary to ensure sustainable resource use and maintain biological balance.

## 5. Conclusions

The results of this research provided evidence that the zooplankton resources in Lake L were more resilient than expected despite being depleted by the stocking of *P. chinensis*. The results suggested that *P. chinensis* could recover from low abundance based on the current

zooplankton resources in the lake. However, the stocking of *P. chinensis* had some impact on the biological balance in the lake, and therefore, long-term and successive investigations are recommended to monitor the effects not only in Lake L but in all lakes stocked with *P. chinensis*. In conclusion, the *P. chinensis* stock could recover in Lake L based on the current zooplankton resources, but ongoing monitoring is necessary.

**Author Contributions:** Conceptualization, Y.H.; methodology, Z.L.; formal analysis, Z.L.; investigation, Z.L., H.Z. and Y.Z.; data curation, Z.L.; writing—original draft preparation, Z.L.; writing—review and editing, F.T.; funding acquisition, F.T. All authors have read and agreed to the published version of the manuscript.

**Funding:** This research was funded by "Project of Research and Development on Applied Technology of Heilongjiang Province", grant number (GA20B202), "The Central-level Non-profit Scientific Research Institutes Special Fund of China, grant number 2020TD07 and HSY201806M", and "Project serving for the Ministry of Agriculture and Rural Affairs of China, grant number A120401".

**Data Availability Statement:** The authors confirm that the data supporting the findings of this study are available within the article.

**Acknowledgments:** We express our sincere gratitude to Qijiapao Fisheries Co., Ltd. and her workers for the support in sampling.

**Conflicts of Interest:** The authors declare no conflict of interest.

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
