# Peer review of "The Resource Status of Plankton after Stocked Protosalanx chinensis Population Collapse in a Lake of Northeastern China"

_water, doi:10.3390/w15101854_

Round 1
Reviewer 1 Report
Comments and suggestion for the authors are in the attached file and in the yellow boxes throughout the text.

Small quality problems in English, mainly typing issues, are indicated throughout the manuscript sent in PDF with text boxes marked in yellow.
Reviewer 2 Report
Review for the paper "The resources status of Plankton after stocked Protosalanx chinensis population collapse in a lake of northeastern China" by Zhe Li, Ying Han, Fujiang Tang, Haoyu Zeng, Yi Zheng submitted to "Water".
General comment.
The introduction of non-native species is one of the most challenging issues in modern studies. Alien species have been found to be responsible for a variety of impacts on native communities. In some cases, community changes are clearly evident through loss of biodiversity, suppression of common taxa, and overall decline in productivity. However, there are also cases of successful introduction of non-native species. The authors conducted plankton surveys in two lakes to assess the consequences of the invasion of a fish (Protosalanx chinensis) on the composition, abundance and biomass of phytoplankton and zooplankton. They found no significant differences in zooplankton abundance and biomass between the lake where Protosalanx chinensis occurred and the lake where this fish was absent. However, the authors found a significant correlation between fish abundance and a specific group of zooplankton (Cladocera), suggesting top-down control of the cladoceran population through ingestion by the fish. The study is based on a comprehensive data set. Samples were collected and handled according to standard procedures. Statistical processing is generally relevant. However, there are many important concerns that need to be addressed before a final decision on the paper can be made. The current MS is not suitable for publication and should be revised according to my suggestions.
Major concerns.
Introduction.
1. Much attention should be paid to the role of non-native taxa in the reorganization of natural ecosystems. Therefore, a brief overview of successful and unsuccessful introductions should be provided.
2. The authors must emphasize the importance and novelty of the study focusing on ecological aspects of biological invasions and introductions of alien species.
Materials and Methods.
1. A brief description of the environmental background would improve the quality of the paper. Provide data on climatic conditions, lake area, main human activities, etc.)
2. Procedures for processing phyto- and zooplankton must be carefully described. Provide data on identification (relevant guidelines), enumeration, calculation of microalgal and zooplankton abundance, biomass calculations (give references and units for phyto- and zooplankton biomass).
3. Describe in detail the fish handling procedures (age determination, measurements, estimation of total density, etc.).
4. Provide a table with site information for each sampling event (date, season, location, etc.). Indicate the total number of plankton samples and fish caught in each sampling event.
Results.
1. The authors presented only combined data for all seasons, whereas it would be good to present data for each season separately and compare plankton assemblages for each period.
2. Similarly, the different seasons need to be compared in terms of dominant species of phyto- and zooplankton as well as diversity indices.
3. Results. Indicate significance levels for all comparisons (between sites, seasons, lakes, and plankton assemblages).
4. Authors must provide data on biological variables of the fish caught for each sampling season (abundance, length, age structure). There is also no mention regarding the status of Protosalanx chinensis in Lake Longhu. The authors only stated a population collapse, but did not provide any evidence to support this.
5. Section 3.3. The authors correlated zooplankton/phytoplankton density with fish length. However, to assess the potential impact of fish on total zooplankton/phytoplankton, they would correlate fish density and zooplankton/phytoplankton abundance. Therefore, I suggest that this analysis be included in the MS.
Discussion.
1. The results regarding seasonal changes in phytoplankton/zooplankton composition, abundance and biomass must be carefully discussed and compared with other studies or similar regions.
2. The effects on fish must be clearly validated by comparison with other reports.
Conclusions must be expanded, focusing on the ecological significance of the authors' results. Potential applications of the authors' data should be highlighted.
Specific remarks.
L14. Consider replacing "Lake Q, P. Chinensis free" with "Lake Q, P. chinensis free".
L15. Consider replacing "though" with "through ".
L17-18 and below in the text (Results). I recommend rounding the abundance and biomass estimates to the nearest whole number for better presentation (e.g. 992*104 ind/L, 9 mg/L).
L18. Indicate the units for zooplankton biomass (wet, dry, carbon).
L25. Consider replacing "And long-term" with "Long-term".
L46. Consider replacing "P. chinensis (Abbott 1901)" with "Protosalanx chinensis (Abbott 1901)".
L47. Consider replacing "And zooplankton is the main food" with "Zooplankton are the main foods".
L63. Consider replacing "are shallow alkaline lake" with "are shallow alkaline lakes".
L74. Grab is used to collect benthic samples. May be: Bottles were used…? Or delete grab in L74 and L76.
L132. Consider replacing "Phytoplankton species composition" with "Phytoplankton composition".
L167. Consider replacing "And Lake L had" with "Lake L had".
L177. Consider replacing "( Figure 3)." with "(Figure 3).".
L180. Consider replacing "October. (0.03)" with "October (0.03)".
L223. Consider replacing "datum" with "data".
L224. Consider replacing "Test" with "Tests".
L225. It is unclear what does mean the total diversity index. The authors calculated H, J, and D only.
Figure 4 is not informative and must be presented in a different way.
L300. Consider replacing "Zooplankton resources not being restrictive on P. Chinensis reviving in Lake L" with "Zooplankton response on P. chinensis presence in Lake L".
L301. Consider replacing "in the larvae" with "at the larval".
L302. Consider replacing "Cyciopidae" with "Cyclopoida".
L302. Bosmina must be in Italic.
L314. Consider replacing "reviving" with "inhabiting".
L325-326. Daphnia, Bosmina, Chydorus must be in Italic.
I have found no reference to Fig. S1 in the main text. Provide this or delete the Supplement.
Literature.
In some cases, the Latin names of the species or genus are not italicized (e.g. Microcystis L427, Oncorhynchus nerka L 452). I recommend careful checking the cited literature.
The English is of low quality.
Round 2
Reviewer 2 Report
The authors have revised the paper according to my comments.
Minor revisions.